# Comparison of Methods for Short-Segment Posterior Stabilization of Lumbar Spine Fractures and Thoracolumbar Junction

**DOI:** 10.3390/jcm13237318

**Published:** 2024-12-02

**Authors:** Agnieszka Tomczyk-Warunek, Michał Kłapeć, Rudolf Blicharski, Sławomir Dresler, Ireneusz Sowa, Andrea Weronika Gieleta, Tomasz Skrzypek, Magdalena Lis, Waldemar Kazimierczak, Tomasz Blicharski

**Affiliations:** 1Laboratory of Locomotor Systems Research, Department of Traumatology, Orthopedics and Rehabilitation, Medical University of Lublin, 20-954 Lublin, Poland; agnieszka.tomczyk-warunek@umlub.pl; 2Clinic of Rehabilitation, Department of Traumatology, Orthopedics and Rehabilitation, Medical University of Lublin, 20-954 Lublin, Poland; rehortop@umlub.pl (M.K.); tomasz.blicharski@umlub.pl (T.B.); 3Department of Clinical Physiotherapy, Medical University of Lublin, 20-954 Lublin, Poland; 4Department of Analytical Chemistry, Medical University of Lublin, 20-093 Lublin, Poland; slawomir.dresler@umlub.pl (S.D.); ireneusz.sowa@umlub.pl (I.S.); 5Department of Plant Physiology and Biophysics, Institute of Biological Science, Maria Curie-Skłodowska University, 20-033 Lublin, Poland; 6Faculty of Medicine and Surgery, University of Malta, Mater Dei Hospital Tal Qroqq, Block A, 2090 Msida, Malta; andrea.gieleta.15@um.edu.mt; 7Department of Biomedicine and Environmental Research, Faculty of Medicine, John Paul II Catholic University of Lublin, 20-708 Lublin, Poland; tomasz.skrzypek@kul.pl (T.S.); magdalena.lis@kul.pl (M.L.); waldemar.kazimierczak@kul.pl (W.K.)

**Keywords:** thoracolumbar junction, lumbar spine, injury, short-segment stabilization, Cobb angle, VAS, ODI, anterior column height

## Abstract

**Background**: Thoracolumbar and lumbar spine injuries account for 30–60% of spinal fractures, especially at the thoracolumbar junction. Conservative treatment is recommended for stable fractures without neurological symptoms, but studies suggest surgical intervention may offer better outcomes. However, there is no consensus on the best stabilization method. **Methods**: This non-randomized, prospective study was conducted on 114 patients divided into groups based on the surgical technique selected: pedicle stabilization using Schanz screw constructs (Group One, n = 37) stabilization above and below the fractured vertebra using pedicle screws (Group Two, n = 32), and intermediate fixation with a pedicle screw additionally inserted into the fractured vertebra (Group Three, n = 45). Outcomes were assessed using the Cobb angle, anterior and posterior vertebral wall height, and patient quality of life via the Visual Analog Scale (VAS) and Oswestry Disability Index (ODI). X-ray imaging was performed before, during, and after surgery in the control group. **Results**: This statistical study showed that the location and type of injury significantly influenced the choice of short-segment stabilization method. In the case of measuring the Cobb angle and the high anterior wall, the statistical analysis showed that the best result was observed in the Schanz Group. Patients from this study group had the lowest pain and the highest efficiency. **Conclusions:** Schanz screw stabilization may offer superior outcomes for thoracolumbar spine injuries, providing better clinical and quality of life results compared to other methods.

## 1. Introduction

Some of the most common injuries of the entire spinal column are injuries of the lumbar spine and thoracolumbar junction, which account for as much as 30 to even 60% of all spinal fractures [1,2]. Among all injuries of the thoracic and lumbar spine, as much as 50–60% are fractures of the thoracolumbar junction. Fractures of the thoracic column of the spine account for 25–40%. Fractures of the lumbar spine and the sacrum account for 10–14% of all injuries in this area of the spine [1,2]. Differences in the frequency of thoracic and lumbar spine injuries and their combinations are related to differences in the anatomical structure of these two parts of the spine, which contribute to changes in their biomechanics. The thoracic spine is characterized by greater stiffness. On the other hand, the lumbar spine is much more flexible, which is related, among other things, to the lack of support by the chest, and the intervertebral discs in this part of the spine are thicker. Also, at the junction of the thoracic column with the lumbar column, the orientation of the facet joints changes from frontal-horizontal to sagittal-vertical. It is the transition from the less mobile thoracic spine to the more dynamic structure of the lumbar spine that contributes to intensive changes in the biomechanics of the connection of these two sections, resulting in greater susceptibility of this area of the spine column to fractures [1,2].

The treatment of fractures of the lumbar spine and the thoracolumbar junction is aimed at stabilizing the spinal column, preventing the displacement of damaged structures, protecting against spinal cord injuries, and enabling patients to recover faster [3,4]. One of the methods of treatment is surgical treatment, which, compared to non-surgical treatment, contributes to faster and better stabilization of the fracture and allows immediate mobilization of the patient and early rehabilitation. This method also enables decompression of the spinal canal, increasing the patient’s chances of recovery [2]. Three surgical methods are used in the surgical treatment of lumbar spine and thoracolumbar joint injuries: anterior, posterior, and combined approaches [1,2,3]. The choice of surgical method depends primarily on the type of injury [5]. In the treatment of thoracolumbar junction injuries, minimally invasive methods, which consist in percutaneous stabilization of the fracture, can also be used [1].

An additional technique for severe fractures, often overlooked, is corpectomy. In this procedure, the fractured vertebra is removed and replaced with an expandable cage. This cage restores vertebral height and supports alignment, providing stabilization in cases where other surgical approaches may not be sufficient. The expandable cage can adjust to the patient’s specific anatomical needs, promoting better recovery outcomes and spinal alignment stability.

The most used surgical method is the posterior approach, due to the ease of anastomosis, reduction in bleeding, and a small incision area. In the case of posterior access, short-segment and long-segment stabilization can be used. The advantages of long-segment stabilization include permanent fixation of damaged vertebrae and better stabilization of the spinal canal. On the other hand, this anastomosis contributes to significant immobilization of the spine [4,6]. Hence, short-segment stabilization has become popular in recent years. However, like any method, it has its disadvantages, including deepening of kyphosis, pain, and more frequent destabilization of the fixation [3,4].

In short-segmental stabilization, three different methods are used for the transphyseal fixation of fractures of the lumbar spine and thoracolumbar junction Th12-L1. The first method involves stabilizing one segment above and below the fracture level. The second method involves additional stabilization of the fractured vertebra using indirect Schanz screws, which are specialized screws that provide strong anchorage for external fixation constructs, known as Schanz constructs. These constructs enable the distribution of forces along the stabilized segments, helping to maintain vertebral alignment [7]. The third short-section method involves the use of Schanz screws for pedicle stabilization [7]. It should be noted that, in the currently practiced spine surgery, the aim is not only to stabilize the injuries; the patient’s satisfaction with the performed surgery is expected by eliminating pain and improving health. Currently, many tools are used to assess the quality of life, including questionnaires that allow assessing whether the surgery performed brought a clinical effect and improved the patient’s life [8].

In recent years, there has been a discussion about the best treatment methods for lumbar spine and thoracolumbar junction injuries. However, the available literature is mostly based on retrospective studies, and there are few prospective case reports [2]. Additionally, in the available literature, there is little information comparing these three methods of pedicle stabilization used in patients with fractures of the lumbar spine and the thoracolumbar junction in terms of clinical and radiological assessment. Similarly, there are few results regarding the quality of life of patients after this type of surgery.

## 2. Materials and Methods

### 2.1. Place of Study and Bioethical Committee

This prospective, non-randomized study was conducted at the Independent Public Clinical Hospital No. 4 in Lublin (SPSK4) and included patients operated on in 2013–2019.

The prospective study was approved by the Bioethics Committee at the Medical University of Lublin, al. Racławickie 1 (number KE-0254/68/2019).

### 2.2. Criteria for the Inclusion/Exclusion of Patients from the Prospective Study

Patient inclusion criteria: (a) first surgery; (b) fracture of one motion segment at the thoracolumbar junction or the lumbar spine; (c) AO Spine Classification: A1, A2, A3, A4, and B1; (d) no damage to the spinal cord and no neurological symptoms; (e) absence of skull and chest injuries; (f) patient’s consent to participate in the prospective study. Criteria for exclusion of patients from this study: (a) subsequent surgery; (b) injury to more than one motion segment; (c) vertebral fractures in the area of the cervical spine or the upper thoracic spine; (d) sacrum injuries; (e) injuries of the thoracolumbar junction or lumbar spine classified as types A0 and C; (f) previous spinal injuries; (g) history of osteoarthritis; (h) cancer; (i) high patient burden associated with comorbidities.

### 2.3. Surgical Techniques and Procedures

#### 2.3.1. Patient

Based on the above-mentioned criteria, a total of 114 patients were qualified for this study. The study group of patients underwent short-segmental stabilization from the posterior approach using three surgical techniques: pedicle stabilization using Schanz screw constructs (Group One, n = 37) stabilization above and below the fractured vertebra using pedicle screws (Group Two, n = 32), and intermediate fixation with a pedicle screw additionally inserted into the fractured vertebra (Group Three, n = 45). All surgical procedures were performed by the same operating team.

#### 2.3.2. Description of the Surgical Procedure

Before the operation, all patients underwent standard imaging diagnostics, which included X-ray examination in the lateral and anterior-posterior projection (AP) and CT (computed tomography).

The posterior open midline approach was used with appropriate patient preparation. The operations were performed under general anesthesia. The patient was placed on the stomach, washed, and covered with a sterile operating field. The operated site was determined by X-ray in two projections (AP and Lateral). The skin incision was made above the spinous processes, which had previously been marked with a marker under the control of the X-ray monitor. Cutting above the spinous processes allowed their exposure. Then, after exposing the deeper layers under the skin and above the spinous processes, the thoracolumbar fascia was cut [9]. Then, the paraspinal muscles were dissected and slid off the spinous processes, the laminae of the arches, and the joint capsules, exposing the spinous processes and the arches [9]. The next stage consisted in short-segmental stabilization of the fractured vertebra using one of the three surgical techniques described below.

Group One (n = 37)—pedicle stabilization using Schanz screw construct. The locations of implantation of Schanz screws were determined under X-ray control. The screws were then inserted into the shafts above and below the fracture level. All the steps were repeated on both sides. The Schanz screws were inserted into the cranial and caudal vertebrae to the fracture level on both sides, using the starting point and trajectory described for pedicle screws. In this technique, it is advisable to use the same sizes of Schanz screws in the same shanks. Then, the appropriate length of the beam was adjusted and the whole thing was connected with blockers. The last element after the X-ray inspection was the breaking of the fragments protruding beyond the structure, which were previously used for reposition and distraction (Figure 1).

Group Two (n = 32)—stabilization above and below the fractured vertebra using pedicle screws. The entire surgical procedure was performed under the control of the X-ray apparatus. In this technique, stabilizing pedicle screws were introduced under the supervision of X-ray into healthy shafts above and below the fracture level. The pedicle screws were inserted into the cranial and caudal vertebrae to the level of the fracture on both sides. The screws were connected with rods of a properly selected size in terms of length (Figure 1).

Group Three (n = 45)—intermediate fixation with a pedicle screw additionally inserted into the fractured vertebra. In this X-ray-guided short-segment stabilization technique, pedicle fixation screws were inserted into the healthy shaft above and below the injury. Pedicle screws were inserted into the cranial and caudal vertebrae to the fracture level on both sides. Intermediate screws were inserted into the fractured vertebra. This procedure was performed on both sides of the vertebra. Then, the bolts were connected with beams of a properly selected size in terms of length (Figure 1).

At this stage, regardless of the group, all the patients under the control of the X-ray monitor were assessed as to whether spinal cord decompression was needed. When indirect decompression turned out to be insufficient, direct decompression, i.e., laminectomy, was performed. This procedure was carried out simultaneously on both sides. Also, at the end of the operation, homeostasis, stabilization, and layered wound closure were checked in all the patients. The wound was covered with a dressing.

#### 2.3.3. Postoperative Care

Weeks following the surgery, physiotherapy played a crucial role in the patient’s recovery [10,11]. During the initial six weeks, the patient performed light activities to improve circulation and prevent stiffness, while protecting the surgical site. Range-of-motion exercises for the hips and lower extremities were included, along with short, supervised walking sessions. Core engagement exercises were cautiously introduced to maintain stability.

In the later weeks, the rehabilitation program advanced to core strength, flexibility, and endurance exercises. The patient engaged in light strengthening activities for the back, abdominal, and pelvic muscles while gradually increasing walking distances and participating in controlled, low-impact activities like swimming or stationary cycling.

After 12 weeks, the focus shifted to restoring full function and preparing the patient for daily activities and work. Advanced strengthening and flexibility exercises were implemented, emphasizing posture correction and proper body mechanics to protect the spine during movements. The patient avoided high-impact activities or heavy lifting until cleared by the surgeon.

Throughout the program, exercises were tailored to the patient’s progress, ensuring appropriate intensity and avoiding excessive strain on the spine. Education on proper movement techniques and lifestyle adjustments was provided to support long-term recovery and prevent reinjury.

#### 2.3.4. Radiological Evaluation

Imaging examinations of thoracolumbar junction and lumbar spine injuries were performed using computed tomography (CT) and X-ray images. In the work, CT was performed twice before the procedure to assess the location and degree of damage to the lumbar spine and the thoracolumbar junction. In this study, it was used to classify the injury according to the AO Spine scale. The AO Spine Classification divides spinal injuries into three primary types: Type A (Compression Injuries): these involve compression fractures without significant disruption to the posterior elements of the spine. Type B (Distraction Injuries): these injuries involve the posterior ligamentous complex and often result from flexion-distraction forces. Type C (Translation/Rotational Injuries): these are the most severe injuries and involve significant rotational or translational displacement, resulting in spinal instability [12]. This examination was also performed once after the surgery to assess the degree of fracture stabilization at least one year after the surgery.

On the basis of the obtained X-ray images, the size of the Cobb angle at the injury site, the height of the anterior and posterior vertebral walls, and the ratio of the height of the anterior to posterior wall were calculated.

These measurements were taken before the procedure, immediately after the procedure, and in a follow-up examination carried out at least one year after the operation.

#### 2.3.5. Evaluation of the Quality of Life of Patients After Short-Segment Stabilization

The assessment of the patient’s quality of life after short-segmental stabilization of the fracture of the thoracolumbar junction and the lumbar spine was carried out in a follow-up examination carried out at least one year after the operation. A total of 108 patients participated in this part of the prospective study (6 patients missed the follow-up or died). The Visual Analog Scale (VAS) and Oswestry Disability Index (ODI) questionnaires were used to assess the patient’s quality of life.

### 2.4. Methodology of Statistical Calculations

#### 2.4.1. Data Preparation

Before starting statistical calculations, the data were properly prepared. All gross errors were analyzed and corrected or removed. Then, the data were analyzed to check their normality. All statistical analyses were performed using the statistical program Statistica version 13.3 (TIBCO).

#### 2.4.2. Assessment of Baseline Group Characteristics

To evaluate baseline differences between groups, several variables were compared, including sex distribution, hospital stay duration, time to surgery, neurological symptoms, presence of laminectomy, and injury cause (primarily motor vehicle accidents or falls). These variables were analyzed using chi-square tests for categorical data and Tukey’s HSD test for continuous data. This ensured that the groups were comparable at baseline and any differences that might influence the outcomes were identified.

#### 2.4.3. Data Normality Check

The Shapiro–Wilk test (*p* < 0.05) was used to check whether the data were normally distributed. The results of this test showed that some of the data were not normally distributed. Therefore, some of the statistical calculations were performed using abnormal tests.

#### 2.4.4. ANOVA Calculations and Tukey’s Test for Normal Data

For data that showed normality, an analysis of variance (ANOVA) was performed to compare mean values between different groups. If the ANOVA result was significant, Tukey’s test was performed to determine which groups were significantly different from each other.

#### 2.4.5. Chi-Square Test for Nominal Data

For nominal data, a chi-square test was performed to see if there were significant differences between the groups. For data that were not normally distributed, the Kruskal–Wallis test, which is a non-parametric equivalent of the ANOVA test, was performed. This test allows comparison of medians between different groups.

## 3. Results

### 3.1. General Description of the Patients Included in the Study

#### 3.1.1. Patient Demographics: Age and Sex

This study included 82 (71.9%) men and 32 (28.1%) women. The mean age of the patients in this study was 44 years (Table 1).

#### 3.1.2. Time of Hospitalization and Waiting for Surgery

The average hospitalization time was 11 days. The mean waiting time for surgery was 3 days (Table 1).

#### 3.1.3. Causes of Injury

Falls were the most common cause of the thoracolumbar and lumbar spine injuries. In the examined group of patients, falls contributed to 69 injuries (61.1%). The second most common cause of vertebral injuries was car accidents, which contributed to 40 fractures (35.4%). In two patients (1.77%), the injury occurred as a result of a motorcycle accident. In one patient (0.88%), the vertebral fracture resulted from crushing. In one patient (0.88%), the injury resulted from a helicopter accident (Table 1).

#### 3.1.4. Fracture Location

The injury most often affected L1 and appeared in 60 patients (52.6%). In 23 patients (20.2%), the injury involved L2. L3 was fractured in 14 patients (12.3%). In 13 cases (11.4%), Th12 was fractured. The injury involved L4 in only only patients (3.51%) (Table 1).

#### 3.1.5. AO Spine Classification

The most common type of injury was A2, which was recorded in 46 cases (40.4%). The second most common type of injury was A3, which was diagnosed in 44 patients (38.56%). In 17 patients (14.9%), the spinal injuries were classified as A4. In four cases (3.51%), the vertebral fracture was classified as A1. The least common type of fracture in the study group was B2, which was diagnosed in three patients (2.64%) (Table 1).

#### 3.1.6. Neurological Symptoms and Laminectomy

Five patients (4.49%) had neurological symptoms. However, laminectomy was performed in 38 patients included in this study (33.3%) (Table 1).

#### 3.1.7. Accompanying Injuries

In the study group of patients, as many as 30 additional injuries were diagnosed. The most common accompanying injuries were lower limb injuries (8.77%). The second most common injury was head injury, which occurred in seven patients (6.17%). In four patients (3.51%), upper limb injuries were an additional injury. Additional lumbar spine injuries were diagnosed in three patients 3 (2.64). Two patients (1.77) were additionally diagnosed with cervical spine injuries. Similarly, in two cases (1.77), the accompanying injury was a thoracic spine injury. Multiple organ and abdominal trauma occurred in two patients (1.77) (Table 1).

### 3.2. Description of Patients in Relation to the Chosen Technique of Stabilization of the Injury

#### 3.2.1. Patient Demographics and Surgical Technique

The mean age of patients in Group One, Group Two, and Group Three did not differ statistically significantly and was 43, 44, and 46 years, respectively (Table 2).

As for the influence of sex on the choice of stabilization fixation, the statistical analysis did not show any differences between the study groups. However, when it comes to the percentage share in individual groups, in the case of men, it was observed that the most numerous group was Group Three, which constituted 30.2% (35). The second most frequently chosen method in men was stabilization of the fracture with the method in which Schanz arrowheads are used to stabilize the injury: 21.1% (24)—Group One. The smallest group among men was Group Two 20.2% (23). In the case of women, the largest group was the former Group One, which constituted 11.4% of all the respondents (13). The second largest was Group Three 8.77% (10), and the least numerous, as in the case of men, was Group Two (7.89%; 9) (Table 2).

#### 3.2.2. Time of Hospitalization and Waiting for the Procedure

Patients from Group One were hospitalized for the longest period of about 15 days. Patients from Group Three stayed in the hospital for the shortest period of about 8 days. Group Two patients were hospitalized for an average of 11 days (Table 2). The statistical analysis showed no effect of days of hospitalization on the choice of vertebral fixation (Table 2).

In the case of waiting time for the procedure, patients from Group One and Three waited the longest. This time was on average more than 3 days. In the case of Group Three, the average waiting time was over 2 days (Table 2). The statistical examination of the collected results did not show a significant effect of the waiting time for surgery on the choice of short-segment stabilization fixation (Table 2).

#### 3.2.3. Cause of Injury

The statistical analysis showed no influence of the cause of the injury on the choice of surgical technique used to fix the injury. Injuries resulting from a fall were most often stabilized in all three study groups, i.e., Group One, Two, and Three: 17.7% (20), 17.7% (20), and 25.7% (29), respectively. Similarly, injuries resulting from a car accident were stabilized with all three fixation methods: 14.16% (17), 8.85% (10), and 12.39% (14), respectively. In two patients (1.77%), whose injury was caused by a motorcycle accident, the vertebral fractures were fixed with stabilization below and above the fracture level (Group Two). In one patient (0.88%), the spinal injury resulted from crushing and was fixed with the technique of indirect stabilization (Group Three). Similarly, the helicopter accident contributed to one injury (0.88%), which was fixed below and above the fracture level (group Two) (Table 2).

#### 3.2.4. Fracture Location

The statistical study showed a significant relationship between the choice of short-segment fixation technique and the location of the fracture (*p* = 0.004). This significance was observed when comparing Group Two to Three (*p* = 0.007) and Group Three to One (*p* = 0.019). On the other hand, the statistical analysis of the collected data did not show any significant differences between Group Two and One (Table 2).

In the case of injuries of the Th12 and L3 vertebrae, Group Three was the most numerous, as it accounted for 7.89% (9) and 7.02% (8) of all cases, respectively. The Th12 and L3 fractures were also stabilized using the other two short-segment fixation techniques. However, the number of patients who had these techniques performed was significantly smaller. In the case of Th12 vertebra injury, Group One consisted of two patients (1.77%) of the entire group. The same frequency of stabilization of the Th12 vertebra was performed as in Group Two. In the case of stabilization of the L3 injury, the number of patients in Groups One and Two was the same and amounted to 2.63% (Table 2).

In the case of the L1 fracture, Group Three included 24 patients, which accounted for 21.05% of all cases. Twenty-one patients, 18.42%, were qualified to Group One. In this location of injury, the smallest group was Group Two, which included 15 patients (13.16%) (Table 2).

In the L2 fracture case, Group One was the largest, as it included 11 patients (9.65%). The second most frequently chosen method of stabilization of the L2 vertebra injury was the technique below and above the fracture level—Group Two. This technique was used in 7.02% of the subjects (eight patients). The least frequently chosen technique in the stabilization of this injury was the method with the use of intermediate screws—Group Three. It was selected in 3.51% of injuries (four patients) (Table 2).

The L4 fracture was the least common injury in the study group. This injury was diagnosed in four patients, which accounted for 3.51% of all examined cases. Patients with an L4 injury were included in Group Two only (Table 2).

#### 3.2.5. AO Spine Classification

The statistical study showed a significant relationship between the OA injury classification and the choice of fracture stabilization technique (*p* < 0.001). Statistical differences were also observed when comparing the research groups. The comparison of Groups Two and Three revealed statistical significance at the level of *p* < 0.001. For Group One and Group Two, the *p*-value was 0.001. However, between Group One and Group Three, the *p*-value was 0.002.

Injuries classified as A2 were the most common in the study group of patients. Most often, patients with this type of injury were classified into Group Three, which comprised 22.8% of cases (26 patients). The second largest group was Group Two, i.e., 11.40% (13 patients). The least frequent in this type of injury were patients qualified to Group One—6.14% (seven) of the examined cases.

The second most common type of injury according to the AO classification was A3. In this type of injury, the largest group was Group One, which accounted for 21.05% of patients (24). The second group was Group Three with 11.40% of injuries (13). The least numerous group was Group Two with 6.14% (seven) of patients.

Fracture type A4 was most often stabilized in Group Two and accounted for 10.53% (12) of all patients. Group One included four patients, which constituted 3.51% of the study group. One patient (0.88%) was qualified to Group Three.

In the case of fractures A1 and B2, Group Three comprised the largest number of patients: 2.63% (three) and 1.75% (two), respectively. Only one patient was qualified to Group One, which constituted 0.88% of the patients.

#### 3.2.6. Neurological Symptoms

The statistical analysis showed no influence of the occurrence of neurological symptoms in patients on the choice of short-segment stabilization method. Classified into Group Two were 2.63% (three) of patients who developed neurological symptoms. On the other hand, 1.75% (two) of patients with neurological symptoms were included in Group Three.

#### 3.2.7. Laminectomy

The statistical analysis showed a significant influence of laminectomy on the choice of fracture stabilization technique (*p* = 0.034). Statistical differences were observed between Groups Two and Three (*p* = 0.012).

In the case of Group Two, laminectomy was performed most often, in 13.16% (15) patients. However, decompression was performed the least frequently in patients from Group Three—7.89% (nine) cases. Group One did not differ statistically significantly from the other groups and amounted to 12.28% (14) of patients.

### 3.3. Cobb Angle

The statistical analysis showed that, regardless of the short-segment stabilization technique used, the size of the Cobb angle because of fixation was significantly reduced compared to its value before the surgery. This effect was observed both immediately after the surgery and in the follow-up examination carried out at least one year after the surgery. The comparison of the value of the Cobb angle immediately after the operation with its value in the follow-up examination one year after the operation showed that it was significantly deepened in Group One and Group Three. However, these differences were not observed in patients from Group Two (Table 3).

The statistical analysis showed that the size of the Cobb angle had a significant impact on the choice of surgical technique. Patients with the highest value of Cobb angle were stabilized using Schanz screws—Group One. These differences were observed for both Groups Two and Three, while no significant differences were observed between Groups Two and Three (Figure 1, Table 3). In the case of the terms immediately after the surgery and one year after the surgery, no significant differences in the size of the Cobb angle were observed when assessing the impact of the method of stabilizing the injury on the deepening of kyphosis.

### 3.4. Vertebral Height

The statistical analysis showed that, regardless of the chosen short-segment stabilization technique, there was a significant difference in the AVH and PVH value when comparing the height before the surgery with the result obtained after the surgery. It should be noted that the effect of improving AVH was maintained only in Group One. In the other research groups, the column height was reduced, and there were no statistically significant differences between the result after the operation and the result obtained in the control examination. The reduction in PVH in the follow-up examination was observed in all study groups.

The comparison of the study groups with each other showed that AVH and PVH before the operation were significantly higher in Group Three compared to Group Two (Table 4). In the case of AVH, a significantly higher value was also observed both in the examination performed immediately after the surgery and in the follow-up examination. However, PVH was significantly higher only in the measurement made immediately after the surgery.

### 3.5. Assessment of Patients’ Quality of Life

#### 3.5.1. VAS—Visual Analog Scale

Figure 2 presents the results regarding the constant pain perception by patients on the VAS, depending on the technique used to stabilize the vertebral fracture, at least one year after the surgery.

The statistical analysis of the responses shows that the average pain perception by patients from Group One was significantly lower than in the other study groups. There were no significant differences in the level of pain perception between Groups Two and Three at least one year after the surgery.

#### 3.5.2. ODI—The Oswestry Disability Index

In Figure 3, the ODI scores for each group are presented. The statistical analysis showed statistically significant differences between the study groups. The lowest degree of disability was observed in patients whose injury was stabilized with the use of Schanz screws (Group One). On the other hand, the highest ODI was recorded in patients whose injury was fixed with pedicle screws above and below the fractured vertebra.

## 4. Discussion

The primary objective in treating injuries to the lumbar spine and thoracolumbar junction is to achieve a stable and lasting reduction in the fracture, as this stability is crucial for promoting the patient’s rapid mobilization and return to functional activity. By securely stabilizing the fracture, early mobilization can be achieved, reducing the risks associated with prolonged immobility, such as muscle atrophy, decreased cardiovascular conditioning, and potential complications like pressure ulcers. Restoring the normal sagittal alignment of the spine, while essential for long-term functional outcomes and reducing the risk of chronic pain or deformity, is considered a secondary goal. Correct sagittal alignment supports overall spinal health and load distribution, but immediate stability and mobilization take precedence to optimize recovery [13].

The treatment of thoracolumbar junction injuries is questionable, especially in patients without severe neurological symptoms. Conservative treatment is often recommended in this group of patients. However, surgical treatment gives the possibility of improved stabilization and, in the future, better prognosis for the patient [8]. In patients without neurological symptoms, the decision on surgical treatment is made on the basis of the size of post-traumatic kyphosis and the degree of deformity [13]. Currently, in spinal injury surgery, not only is the stabilization and fusion of the fracture sought, but also the aim of this treatment method is to reduce pain and improve patient’s performance. This is why surgical treatment is preferred by doctors [8]. In the surgical method of treatment, the posterior approach is most often used, which is associated with fewer complications compared to the anterior approach, and operators prefer this type of approach due to its easier performance [14]. For the surgical treatment of spinal injuries from the posterior approach, short- and long-segment stabilization is used. The traditional method of treating fractures is long-segment fixation. However, long-sectional stabilization contributes to a significant loss of mobility and increases the risk of degradation of adjacent motion segments [15,16]. It should be noted that the goal of stabilization is not only to fuse the injury and decompress the spinal cord, but a very important aspect in the surgical treatment of spinal injuries is to maintain maximum mobility by reducing the number of segments that have been fused, which is possible thanks to short-segment stabilization.

This type of stabilization consists in the insertion of pedicle screws above and below the fracture level, which means that it covers only one motion segment [17]. This type of fixation is characterized by a shorter procedure time and a lower risk of large blood loss. However, this type of stabilization can only be performed when the injury affects only one vertebra and is not recommended in type C fractures [15,16]. In short-segment stabilization, three methods of fusion of thoracolumbar and lumbar spine injuries are used. The first method involves stabilization above and below the fracture level. In the second method, intermediate screws are additionally used, which are inserted into the fractured vertebra. The third method uses Schanz screws [7].

Traditional methods of short-segment stabilization include the technique of using pedicle screws above and below the fracture level [17]. This technique is known and easy to perform; it stabilizes most fractures and is a very popular method [1]. However, according to the literature, this method is associated with a high risk of implantation failure, pseudoarthrosis, neurological deterioration, insufficient decompression of the spinal cord, incomplete correction of kyphosis, and the need for vacuum removal of the instrumentation [18].

The second technique used in short-segment stabilization is a method that also uses pedicle screws above and below the fracture level, but additionally the screw is inserted into the fractured vertebra. The results of several studies have shown that the use of intermediate screws in the fractured vertebra reduces the risk of deepening kyphosis [19,20,21]. As a result, this method allows preservation of the correct biomechanics of the spine in a better way than the traditional method of short-segment stabilization [18]. The disadvantages of this technique include instrumentation failure, development of pseudoosteoarthritis, insufficient correction of kyphosis, high risk of infection, risk of neurological syndrome, etc. It can also contribute to deterioration of the mobility of the adjacent motion segment [1,13].

The third method of short-segment stabilization consists in the use of Schanz screws to fix the fracture. This method allows correction of significant post-traumatic kyphosis and ensures high stability and stiffness fixation [13]. According to the literature, the use of Schanz screws is better because, due to their design, they allow the load effect on the spine to be transferred to Schanz constructs. This method also reduces the risk of postoperative complications. However, like any method, it has its disadvantages, which include a longer operation time and a risk of increased blood loss [22].

As can be seen from the cited literature, each of the short-segment stabilization methods has its advantages and disadvantages. Based on patient’s results and experience, the doctor decides which of these techniques to use in the treatment of patients with thoracolumbar and lumbar spine injuries. In the available literature, the issue of the choice of short-segment stabilization method is still debatable and numerous studies are conducted on this subject [22].

In the case of comparisons of these three methods of short-segment stabilization, there are studies comparing stabilization above and below the fracture level and intermediate stabilization [18]. Also available are results comparing indirect stabilization with the method using Schanz constructions [22]. There are no studies comparing these three methods together. Therefore, the aim of this study was to conduct a study on a group of patients with injuries of the thoracolumbar joint and the lumbar spine, who underwent short-segmental stabilization using these three methods. In addition, in this study, the quality of life of the patients at least one year after the surgery was assessed.

In the presented study, a higher frequency of thoracolumbar and lumbar spine injuries was observed in men than in women. Men accounted for 71.93% of the study group, and women represented 28.07%. These data are consistent with the available literature, in which men also constitute the largest group of patients with damage to the thoracolumbar and the lumbar spine [1,17]. In a report from China, the ratio of men to women in patients with thoracolumbar junction injuries was 1.4:1 [23,24]. In the discussed results, this ratio was higher and amounted to 2.6:1 (men to women). This higher ratio may have been due to the fact that our study also included lumbar spine injuries and that only patients without neurological deficits were included in this study. The literature has shown that women have a higher tendency to neurological symptoms than men. As well as in Sidon et al. (2018), differences were observed in the location and causes of injuries between women and men; this may be related to the sexual dimorphism in the structure of the skeleton and the muscular system, which significantly affects the different biomechanics of the spine [25].

The average age of patients in the present study was 44 years. In the study conducted by Rajasekaran et al. (2015), the average age of patients with thoracolumbar and lumbar spine injuries was between 20 and 40 years [1]. In the work reported by Hamdan et al. (2021), the mean age of patients was 32 years [26]. The publication by Fernández-de and De (2023) contains information that the age of patients with the most frequently reported injuries of the thoracolumbar junction ranged from 15 to 29 years. At present, the average age is 35 [23]. In a report published by Li et al. (2019), it was shown that the average age of patients was 49 years [24]. In a study conducted by Al Mamun Choudhury et al. (2023), the average age of patients included in a clinical trial in which they underwent short- and long-segment stabilization was 32.30 years in one research group and 33.13 years in the other [17]. Only in the work of Spurgas et al. (2022) was the mean age of patients included in a prospective study with a thoracolumbar junction injury 44.3 years [27]. The cited works showed that the age varies depending on the studied population. The available literature lacks information on the average age of patients with thoracolumbar and lumbar spine injuries in the Polish population. The statistical analysis performed in the present study showed that neither the sex nor the age of the patients qualified for the clinical trial had a significant impact on the choice of short-segment stabilization method.

In the presented work, the average time of hospitalization was 11 days. The waiting time for surgery was 3 days. The statistical analysis showed no influence of the average surgery time and waiting time for the procedure on the choice of short-segment stabilization technique. There is a discussion in the literature regarding the timing of stabilization after injury. The available literature provides the results of the work of Vaccaro et al. (1997), who assessed whether the time from injury to surgery affected neurological and functional outcomes in patients with traumatic cervical spinal cord injury (SCI). This study compared patients who had surgery less than 3 days after the injury occurred with patients who waited more than 5 days for surgery. The results showed no differences between the study groups [28]. However, in the study reported by La Rosa et al. (2004), which was carried out on a large group of patients, early decompression of spinal cord injury has been shown to give significantly better results [29]. Therefore, when patients suffer from spinal cord injury and neurological symptoms progress, it is recommended to stabilize the injury as soon as possible and decompress the spinal cord [29]. In patients without neurological symptoms or with complete neurological deficit, surgery should be performed when the patient’s condition is stable and the patient is prepared for surgery. In the case of a group of patients without neurological deficits, there are no studies that would indicate when such a procedure should be performed and that early stabilization gives a better result [29]. It should be noted that only patients without significant neurological deficits were qualified for the present study. In terms of the division into groups, Groups One and Two had a similar waiting time for the procedure. On the other hand, patients from Group Three waited the shortest for surgery.

The mean hospital stay in the prospective study was 11 days for all patients. In the case of the division into groups, the patients from Group One stayed in the hospital the longest—about 15 days. Patients from Group Three were hospitalized for the shortest period of about 8 days. Patients from Group Two were hospitalized for an average of 11 days. Compared to the results of other works, the average hospital stay in the study conducted by Aoui et al. (2020) was 12.2 days [30].

In the presented work, the data on the cause of the thoracolumbar injury were also collected. The statistical analysis showed no influence of the etiology of the injury on the choice of the method of short-segment stabilization. However, based on the collected epidemiological data, it is possible to characterize the most common causes of thoracolumbar and lumbar spine injuries in the study group of patients.

In the examined group of patients, the most common cause of thoracolumbar and lumbar spine injuries was falls, which accounted for as much as 61.07% of all cases. The second most common cause was car accidents, which contributed to 40 fractures (35.40%). Also, in a small number of patients, the injury resulted from a motorcycle accident (1.77%), crushing (0.88%), and a helicopter accident (0.88%). In an epidemiological study, Li et al. (2019), dealing with the subject of thoracolumbar injuries, also observed that the most common cause of fractures occurring in this region was falls, which accounted for as much as 43.2% [24]. However, a meta-analysis of the available literature on the frequency of thoracolumbar junction injuries carried out by Katsuura et al. in 2016 showed that the most common cause of injuries was car accidents 36.7%, falls 31.7%, motorcycle accidents 10.05%, knockdown 4.083%, and other 9.03% [31]. Also, in another publication from 2023, there is information that car accidents (36.7%) were the most common cause of injury, followed by falls from a height (31.70%) [23]. However, in both these studies, an article in which patients with spinal cord injuries were also qualified to the research groups was included in the analysis, and these injuries most often occur as a result of traffic accidents [32]. However, in the presented study, patients with spinal cord injuries were not included, which may have been responsible for the different results regarding the etiology of the injury.

The data collected during the prospective study presented in this paper show that, in terms of the location of the injury, the fracture most often affected the L1 vertebra and occurred in 60 patients (52.63%). The obtained results are consistent with other studies [31,33]. As for the frequency of fractures of the other vertebrae: T12, L2, L3, and L4 were injured in 11.41%, 20.17%, 12.28%, and 3.51% of the cases, respectively. The incidence of injuries to L1 may be related to its specific anatomical structure and the fact that it is the first vertebra in which the less mobile thoracic spine turns into the more mobile lumbar spine [34]. In the presented study, a statistical analysis was additionally performed to determine a potential significant effect of the location of the injury on the choice of surgical technique. This analysis revealed significant differences between Groups Two and Three (*p* = 0.007) and Three and One (*p* = 0.019). To the best of our knowledge, this is the first prospective study to perform such an analysis.

The presented work also collects information on the most common types of injuries that occurred in the patients included in this study according to the AO Spine classification. The collected epidemiological data showed that, in the examined group of patients, the cleanest type was A2, which accounted for 40.36% of all injuries. The second most common type of injury was A3, with a diagnosis frequency of 38.58%. In the study group of patients, type A4, A1, and B2 were also diagnosed in 14.91%, 3.51%, and 2.64% of patients, respectively.

In a study conducted by Joaquim et al. (2013), which included over 400 patients with injuries to the lumbar spine or the thoracolumbar junction, the frequency of occurrence of injury types according to the AO classification was assessed as well. Some of the patients included in the study were treated conservatively (310 patients), and surgical treatment was used in the remaining patients (148 patients). In the study, it was observed that the most frequently diagnosed type of injury in these patients was type A, which in the group of patients treated conservatively accounted for as much as 98% of the entire study group. In the case of surgically treated patients, type A was diagnosed in as many as 69.5% of patients, type B in 24.3%, and type C in 6% [35]. In our study, type A without a subtype was diagnosed in 97.36% of the patients included. Similarly, as reported by Magerl et al. (1994) in a study on 1445 patients with injuries of the thoracic spine or the lumbar spine, the most common type of injury was type A, which was diagnosed in 66.1% of patients; type B was diagnosed in 14.5%, and type C in 19.4%. In their study, A1 injuries accounted for 34.7% of the total [36]. In the presented prospective study, there is no frequency of occurrence of type C, because it was one of the criteria for excluding the patient from the study. In this type of injury, stabilization with short-section techniques is not recommended.

Injury classification systems are designed to facilitate the doctor’s decision on the type of treatment used, which is confirmed by the results reported in the presented work. The statistical analysis showed that the type of injury significantly affects the choice of short-segment stabilization technique. Statistical differences were observed between all three research groups, which were divided on the basis of the short-segment stabilization method used. Given the data presented in this work and in the literature, it can be concluded that the type significantly affects the choice of treatment method. However, it should be noted that the type of treatment used should be individually selected for a given patient and is largely dependent on the knowledge and experience of the operator [12,37,38,39].

The presented study also collected information on the frequency of laminectomy in the study group of patients. As shown by these data, this procedure was performed in 33.33% of patients. Laminectomies in the study group of patients were performed both in the case of neurological symptoms and to protect the patient against the risk of neurological symptoms after repositioning. In the latter case, the decision to perform laminectomy was made by the operating team during the procedure. The main purpose of laminectomy during short-segment stabilization is to decompress the spinal cord and to reduce the risk of postoperative neurological symptoms [40].

In the present study, a statistical analysis was also conducted to check whether the number of laminectomies performed depends on the techniques of short-segment stabilization. Statistically significant differences were observed in the study between Groups Two and Three, in which stabilization above and below the fracture and fusion with intermediate screws were performed, respectively. In the case of Group Two, laminectomy was performed most often—13.16%. On the other hand, decompression was performed the least frequently in patients in whom intermediate screws were used to stabilize the fracture (7.89%). The group of patients in whom anastomosis was performed with Schanz screws did not differ statistically significantly from the other groups, and the frequency of laminectomy was 12.28%. Between Group One and Group Two, there was no great difference in the number of patients in whom this procedure was performed. It can be concluded that, in a larger group of patients, Group One would also be statistically different from Group Three. However, it should be noted that this is the first prospective study that examined statistical differences in the number of laminectomies performed, depending on the surgical technique.

In the presented study, the value of the Cobb angle was also determined three times: before the operation, immediately after the operation, and in a follow-up examination carried out at least one year after stabilization. The conducted statistical analysis showed that, regardless of the method of short-segment stabilization, the size of the Cobb angle immediately after the surgery was significantly reduced in relation to its value before the surgery. This effect was also observed in the control study. On the other hand, the comparison of the value of the Cobb angle immediately after the operation with its value obtained in the follow-up examination revealed that it was significantly deepened in Group One and Group Three. However, in patients from Group Two, there were no significant differences in the value of the Cobb angle immediately after the surgery and in the follow-up examination.

The statistical analysis showed that the size of the Cobb angle had a significant impact on the choice of surgical technique. In patients with the highest Cobb angle value, stabilization was performed using Schanz screws; these differences were observed between Groups Two and Three. However, no significant differences were observed between Groups Two and Three. As mentioned in the introduction, the technique of short-segment stabilization using Schanz screws allows a significant reduction in the post-traumatic Cobb angle, which is also confirmed by the results obtained in the presented prospective study [13]. In the case of the periods immediately after the surgery and one year after the surgery, there were no significant differences in the size of the Cobb angle between the study groups.

In a study conducted by Ökten et al. (2015), a better correction of the Cobb angle was obtained in indirect fixation compared to fixation where pedicle screws were inserted only above and below the fracture level [3]. Also, the study reported by Kanna et al. (2015) showed a better correction of the Cobb angle in indirect stabilization [41]. Similarly, in a study published in 2010 by Farrokhi et al., there were significant differences in the correction of post-traumatic kyphosis and the size of the Cobb angle in a follow-up examination carried out 6 months after the procedure between the group of patients who were stabilized by inserting pedicle screws above and below the fracture level and patients who underwent indirect fixation. Better results were obtained in patients in whom the method with the use of intermediate screws was used to stabilize the injury [26]. However, in the presented study, no significant differences in the correction of the degree of kyphosis were observed between Groups Two and Three undergoing the traditional method of short-segment stabilization and indirect fusion with the use of pedicle screws in the fractured vertebra, respectively. Some literature results indicate that the traditional technique is burdened with a greater risk of deepening kyphosis and loss of correction [18]. However, when comparing the examined time intervals in the present study, no deepening of the degree of kyphosis in relation to the result obtained immediately after the operation was observed only in Group Two. On the other hand, in the remaining groups, where indirect stabilization and fixation with the use of Schanz screws were performed, an increase in the value of the Cobb angle in the control examination was observed in relation to the result obtained immediately after the operation. However, between the groups, these differences were not statistically significant, which could be related to the small group size.

In a study carried out by Gómez et al. (2021) on patients with injuries of the thoracolumbar junction, in whom fracture stabilization was performed with the use of Schanz screws, the size of the Cobb angle was also assessed [13]. This study showed a significant reduction in the Cobb angle in states of post-traumatic kyphosis. The same results were observed in the presented work. The comparison of the size of the post-traumatic kyphosis with the size of the Cobb angle in the follow-up examination carried out at least one year after the operation demonstrated that the pre-surgery state did not return. However, as shown by the comparison of the results at the time immediately after the operation with those obtained in the control examination, the Cobb angle deepened but did not return to the pre-operation state.

In the discussed study, no significant differences between the study groups were noted, but regardless of the short-segment stabilization technique used, a significant improvement in post-traumatic kyphosis was observed and this effect was maintained in the control study. These findings are consistent with the results published in recent years, in which short-segment fixation was also used to stabilize the injury of the thoracolumbar junction and the lumbar spine. They confirm that this type of stabilization makes it possible to properly adjust the fractured vertebra and restore normal kyphosis, and this effect is maintained in follow-up examinations [42,43,44].

In the presented work, the posterior and anterior vertebral height was also measured. The measurements were made three times: before the operation, immediately after the operation, and in a follow-up examination carried out at least one year after the surgery.

The obtained results show that, regardless of the chosen short-segment stabilization technique, the height of the AVH improved immediately after the operation. However, this effect persisted only in groups where the short-segment stabilization was performed using Schanz screws. It should also be noted that Group Two, in which stabilization above and below the level of the injury was performed, had significantly lower AVH before the operation compared to Group Three. Also, AVH was lower than in Group One, but without statistical significance. This difference may have significantly contributed to the greatest height reduction in the follow-up examination in the patients from this group. However, a study on a larger group of patients is needed to confirm this issue.

Ökten et al. (2015) showed better improvement in AVH in a group in which intermediate stabilization was performed [3]. This was also evident in our work. However, as mentioned earlier, these differences may be due to the fact that Group Two, in which traditional short-segment stabilization was performed, already had the lowest height of the anterior spinal column in the pre-operative examination compared to other groups. Also, a clinical study was performed to compare the improvement of AVH between groups of patients with long-segment fixation and short-segment fixation with intermediate fixation. It showed that intermediate stabilization gives the same effect as long-segment fixation [20].

In our study, the best effect of increasing the height of the anterior spinal column was observed in Group One. In a study carried out by Gómez Vega et al. (2021), an increase in vertebral height after short-segment stabilization using Schanz screws was noted as well. However, their study lacks a second group in which a different type of short-segment stabilization would be performed to compare the results obtained. In our study, the stabilization with Schanz screws was compared to two other short-segment stabilization techniques, and the best effect was still observed in this group. Similarly, in this group of patients, the pre-surgery state was not restored, as in the case with the other patients [13].

In the case of PVH, in all patients, regardless of the group, there was an improvement immediately after the procedure compared to the result obtained before the procedure. However, there were no statistically significant differences between the measurement made before the examination and the result obtained in the follow-up examination carried out at least one year after the operation. In a study reported by Hamdan et al. (2012), the PVH was measured in patients with injuries of the thoracolumbar junction that underwent short-segment stabilization with the use of intermediate screws [26]. It was observed that the effect of height improvement was maintained both after the operation and in the follow-up examination. However, in this study, only percentage improvement was calculated, and statistical significance was not determined.

Based on the results obtained from the measurements of the size of the Cobb and AVH angles, it can be concluded that the best method of short-segment stabilization is fixation with the use of Schanz screws.

In the presented study, a clinical study was also performed, based on patients completing questionnaires regarding the pain sensation scale (VAS) and the disability index (ODI). The obtained results show the best effect in patients from the group in which stabilization was performed with the use of Schanz screws. The presented data show that the best short-segment stabilization technique is fixation with the use of Schanz screws. As neither patients nor evaluators were blinded to the interventions, there is a potential for observational bias, particularly in subjective assessments such as pain and disability scores.

However, our work has a few limitations resulting from the quasi-experimental study design. The first limitation resulting from the study design is the lack of randomization. The quasi-experimental study design and lack of blinding in case assessment in level of pain and disability may impact both generalizability and internal validity. Without randomization, group differences due to the surgical technique or patient characteristics may introduce confounding variables, potentially limiting the study’s ability to establish causal relationships. Although baseline characteristics were statistically similar across groups, unmeasured variables such as patient-specific factors (e.g., bone density, comorbidities), surgeon-related factors (e.g., experience and technical proficiency), and variability in post-operative rehabilitation could have influenced the outcomes. Unfortunately, the scope of this study did not include systematic collection of these data, precluding a formal sensitivity analysis. However, one of the strengths of our study is the rigorous evaluation of baseline characteristics, ensuring that the groups were comparable in terms of sex ratio, hospitalization duration, time to surgery, injury mechanism, and key clinical factors such as the presence of neurological symptoms or laminectomy. These efforts, using robust statistical methods (e.g., Chi-square test, ANOVA, Kruskal–Wallis test), aimed to reduce bias associated with non-randomized group allocation. Nevertheless, despite these measures, unmeasured confounders such as variations in rehabilitation adherence, differences in surgeon proficiency, and patient-specific factors (e.g., bone density, comorbidities) may still have influenced the outcomes. These limitations highlight the need for cautious interpretation of our findings and the importance of future studies employing randomized designs or advanced statistical techniques such as propensity score matching to validate our results.

A second limitation of the study design is the lack of blinding. However, it should be noted that our study could not be blinded because before the procedure, the patient had to sign an informed consent form for the procedure. Before signing it, the patient had to be informed about how the operation would be performed, and obtain information about other treatment options and the risks associated with the surgery. Nevertheless, the absence of blinding increases the likelihood of observational bias, particularly in subjective assessments, which may further affect the internal validity. In our work, informing patients about the treatment method could have influenced their assessment of their quality of life, which in our study was carried out using the WAS and ODI scales. On the other hand, the available literature contains information that blinding is not always possible and the lack of it does not significantly affect the results obtained. Meta-analysis conducted by Moustgaard et al. 2020 and by Efficace et al. 2022, in which blinded and unblinded studies were compared, showed no significant differences in the obtained results [45,46]

However, since no such research exists for assessments of quality of life following short-segment spinal stabilization, we are unable to determine how the lack of blinding might have influenced the results. We therefore recommend that future studies in this area incorporate randomization and blinding procedures to address this limitation.

Another limitation of this study is the absence of control over certain variables in postoperative care that could significantly influence the study outcomes. While the research provides valuable insights, the lack of standardization for variables such as patient adherence to postoperative instructions or differences in individual baseline health conditions limits the reliability of its conclusions. These uncontrolled factors introduce potential confounding effects, making it difficult to isolate the impact of the surgical interventions under investigation. Future studies should aim to incorporate tighter controls and stratified analyses to address these gaps, ensuring more robust and generalizable findings.

In summary, this is the first prospective study to compare three short-segment stabilization techniques with each other. Based on the obtained data, the location of the injury as well as its type has a significant impact on the choice of short-segment stabilization technique. There were also significant differences in the number of laminectomies performed between the study groups. It should be noted that, regardless of the fixation technique chosen, the correct curvature of the spinal column can be significantly restored, although no significant differences between the groups were observed in the postoperative follow-up examination. However, it should be noted that the highest post-traumatic kyphosis was observed in patients in the Schanz group, which proves that this technique is preferred in patients with a large Cobb angle [13]. It was also observed that the Schanz technique ensured a better recovery of AVH. As well as the assessment of the quality of life of patients after the operation, using the WAS and ODI scales showed that the stabilization using Shanz screws is significantly better. Therefore, it can be stated that in medical practice, this may be the best type of short-segment stabilization of lumbar spine and thoracolumbar junction injuries, especially in patients with high fracture kyphotization. However, the lack of analysis of confounding factors and blinding of this study may have influenced the results. Our study was also limited to one hospital, one patient population, and one surgical team performing the procedures. We therefore recommend further studies in this direction to confirm this hypothesis. Our team also recommends further studies in this direction based on blinding and randomization to obtain more reliable results. We also recommend that further studies be conducted in several centers to check whether the obtained results are influenced by the patient and can be reproduced by other surgical teams.

## Figures and Tables

**Figure 1 jcm-13-07318-f001:**
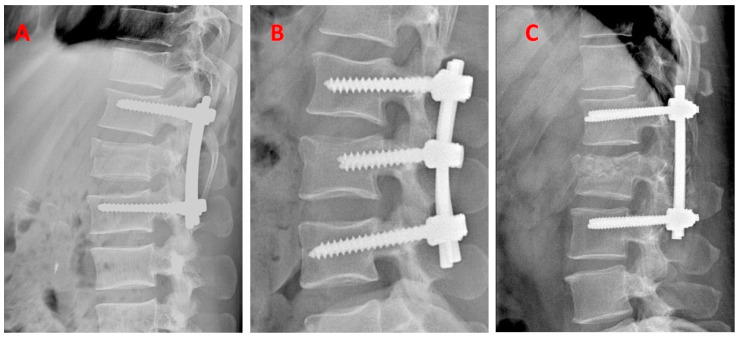
Representative X-ray images of each of the three short-segment stabilization techniques. (**A**)—Group Three—intermediate fixation with a pedicle screw additionally inserted into the fractured vertebra (n = 45); (**B**)—Group Two—stabilization above and below the fractured vertebra using pedicle screws (n = 32); (**C**)—Group One—pedicle stabilization using Schanz screw constructs (n = 37).

**Figure 2 jcm-13-07318-f002:**
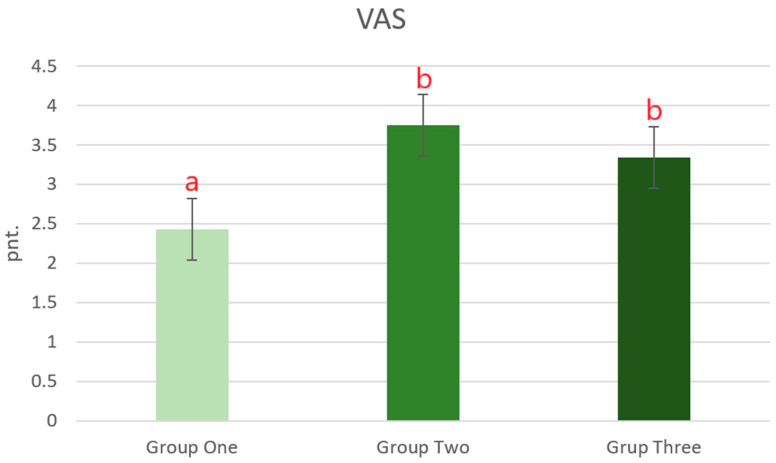
Level of pain perception based on the VAS by patients, depending on the applied stabilization of the fracture of the thoracolumbar vertebra and the lumbar spine using one of the three short-segment stabilization techniques at least one year after the surgery. Group One—pedicle stabilization using Schanz screw constructs (n = 35); Group Two—stabilization above and below the fractured vertebra using pedicle screws (n = 32); Group Three—intermediate fixation with a pedicle screw additionally inserted into the fractured vertebra (n = 41). Data are mean values ± standard error of the mean (whiskers). ^a,b^—significant difference: *p* < 0.05 (Tukey’s HSD test).

**Figure 3 jcm-13-07318-f003:**
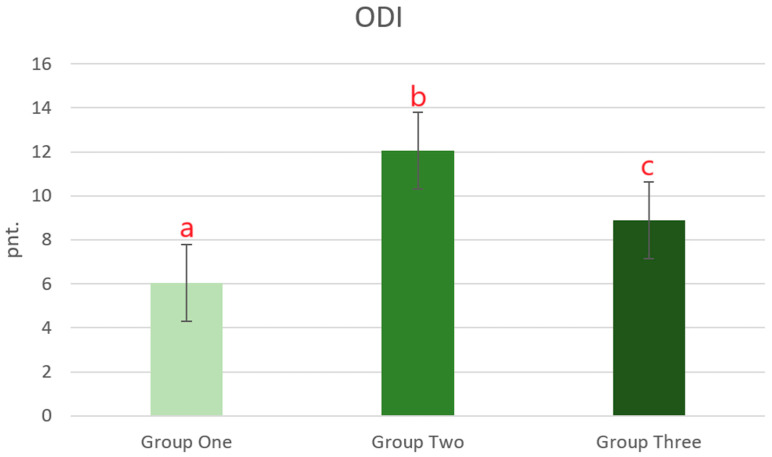
ODI score in patients, depending on the applied stabilization of the fracture of the thoracolumbar junction and the lumbar spine using one of the three short-segment stabilization techniques at least one year after the surgery. Group One—pedicle stabilization using Schanz screw constructs (n = 35); Group Two—stabilization above and below the fractured vertebra using pedicle screws (n = 32); Group Three—intermediate fixation with a pedicle screw additionally inserted into the fractured vertebra (n = 41). Data are mean values ± standard error of the mean (whiskers). ^a,b,c^—significant difference: *p* < 0.05 (Tukey’s HSD test).

**Table 1 jcm-13-07318-t001:** General characteristics of the patients participating in this study.

	N (%)
Sex	
Women	32 (28.1)
Men	82 (71.9)
Age, years	
44 ± 13.7	
Average hospitalization time, days	
11.5 ± 22.0	
Average waiting time for surgery	
3.34 ± 3.47	
Cause of injury, days	
Car accident	40 (35.4)
Fall	69 (61.1)
Motorcycle accident	2 (1.77)
Body crush	1 (0.88)
Helicopter accident	1 (0.88)
Fracture location	
Th12	13 (11.4)
L1	60 (52.6)
L2	23 (20.2)
L3	14 (12.3)
L4	4 (3.51)
AO Spine Classification	
A1	4 (3.51)
A2	46 (40.4)
A3	44 (38.6)
A4	17 (14.9)
B2	3 (2.64)
Neurological symptoms	
Yes	5 (4.39)
Laminectomy	
Yes	38 (33.33)
Accompanying injuries	
Head injuries	7 (6.14)
Lower limb injuries	10 (8.77)
Upper limb injuries	4 (3.51)
Additional fractures of the cervical spine	2 (1.77)
Additional fractures of the thoracic spine	2 (1.77)
Additional fractures of the lumbar spine	3 (2.64)
Multiple organ trauma	1 (0.88)
Abdominal trauma	1 (0.88)

Legend: n-number of patients with short-segment stabilization of the lumbar spine and thoracolumbar junction injury and % of all patients included in this study. Age, mean hospitalization time, mean waiting time for surgery: mean ± standard error of the mean.

**Table 2 jcm-13-07318-t002:** Characteristics of a group of patients with spinal column injuries after short-segment stabilization performed with three techniques. Influence of demographic characteristics, injury location, type of injury, neurological symptoms, and laminectomy on the choice of short-segment stabilization method.

	Group One n = 37 (32.4%)	Group Two n = 32 (28.1%)	Group Three n = 45 (41.2%)	*p **	*p* Group Two-Three	*p* Group Two-One	*p* Group Three-One
Age							
	43.3 ± 13,7	44.2 ± 15.8	46.4 ± 15.7	NS	NS		
Sex							
Woman, % (n)	11.4 (13)	7.89 (9)	8.77 (10)	NS	NS		
Man, % (n)	21.1 (24)	20.2 (23)	30.7 (35)				
Average hospitalization time							
	15 ± 38	11 ± 5	8 ± 3	NS	NS		
Average waiting time for surgery							
	3.66 ± 4.78	3.69 ± 3.18	2.82 ± 2.17	NS	NS		
Cause of injury							
Car accident, % (n)	14.2 (16)	8.85 (10)	12.4 (14)	NS	NS		
Fall, % (n)	17.7 (20)	17.7 (20)	25.7 (29)				
Motorcycle accident, % (n)	0.00 (0)	1.77 (2)	0.00 (0)				
Body crush, % (n)	0.00 (0)	0.00 (0)	0.88 (1)				
Helicopter accident, % (n)	0.88 (1)	0.00 (0)	0.00 (0)				
Fracture location							
Th12, % (n)	1.75 (2)	1.75 (2)	7.89 (9)	*0.004*	*0.007*	*0.154*	*0.019*
L1, % (n)	18.4 (21)	13.2 (15)	21.1 (24)				
L2, % (n)	9.65 (11)	7.02 (8)	3.51 (4)				
L3, % (n)	2.63 (3)	2.63 (3)	7.02 (8)				
L4, % (n)	0.00 (0)	3.51 (4)	0.00 (0)				
AO Spine Classification							
A1, % (n)	0.88 (1)	0.00 (0)	2.63 (3)	*<0.001*	*<0.001*	*0.001*	*0.002*
A2, % (n)	6.14 (7)	11.4 (13)	22.8 (26)				
A3, % (n)	21.1 (24)	6114 (7)	11.4 (13)				
A4, % (n)	3.51 (4)	10.5 (12)	0.88 (1)				
B2, % (n)	0.88 (1)	0.00 (0)	1.75 (2)				
Neurological symptoms							
Yes, % (n)	0.00 (0)	2.63 (3)	1.75 (2)	NS	NS		
Laminectomy							
Yes, % (n)	12.3 (14)	13.2 (15)	7.89 (9)	*0.034*	*0.012*	*0.448*	*0.074*

Legend: n-number of patients with short-segment stabilization of the lumbar spine and thoracolumbar junction injury and % of all patients included in this study. Age, mean hospitalization time, mean waiting time for surgery: mean ± standard error of the mean. Group One—pedicle stabilization using Schanz screw constructs (n = 37); Group Two—stabilization above and below the fractured vertebra using pedicle screws (n = 32); Group Three—intermediate fixation with a pedicle screw additionally inserted into the fractured vertebra (n = 45). Significant difference: *p* < 0.05 (Chi^2 NW test, Tukey’s HSD Test); *p*-presence of differences between the examined groups, *p* * < 0.05—the effect of the group on the value of the examined parameter.

**Table 3 jcm-13-07318-t003:** Evaluation of the Cobb angle before surgery, immediately after surgery, and at least one year after surgery in patients whose thoracolumbar junction and lumbar spine injuries were stabilized using one of the three short-segment fixation techniques.

	Group One	Group Two	Group Three
Preoperative, °	13.2 ± 0.892 ^a^	8.71 ± 0.694 ^b^	8.86 ± 0.706 ^b^
Postoperative (right after the operation), °	3.24 ± 0.704 ^de^	2.58 ± 0.651 ^de^	2.61 ± 0.551 ^e^
Postoperative (at least one year after the operation), °	6.22 ± 0.808 ^bc^	4.68 ± 0.693 ^cde^	5.00 ± 0.745 ^cd^
Average percentage of improvement between preoperative and right postoperative measurements, %	75.5	70.4	70.5
Average percentage of improvement between preoperative and minimum one-year postoperative measurements, %	53.1	46.3	43.6
Average percentage of deepening between right postoperative and one-year postoperative measurements, %	47.8	44.8	47.7

Group One—pedicle stabilization using Schanz screw constructs (n = 37); Group Two—stabilization above and below the fractured vertebra using pedicle screws (n = 32); Group Three—intermediate fixation with a pedicle screw additionally inserted into the fractured vertebra (n = 45). Data are mean values ± standard error of the mean (whiskers). ^a,b,c,d,e^—significant difference: *p* < 0.05 (Tukey’s HSD test). ° It symbolizes cobb angle.

**Table 4 jcm-13-07318-t004:** Anterior and posterior vertebral height before, immediately after, and at least one year after the surgery in patients who had fractures of the thoracolumbar junction, or the lumbar spine stabilized using one of the three short-segment stabilization techniques.

		Group One	Group Two	Group Three
AVH	Preoperative	1.86 ± 0.093 ^cd^	1.74 ± 0.076 ^d^	2.17 ± 0.052 ^bc^
	Postoperative (immediately after the operation)	2.40 ± 0.121 ^ab^	2.10 ± 0.085 ^bc^	2.41 ± 0.067 ^a^
	Postoperative (at least one year after the operation)	2.21 ± 0.118 ^ab^	1.71 ± 0.076 ^d^	2.14 ± 0.079 ^bc^
PVH	Preoperative	2.62 ± 0.053 ^e^	2.73 ± 0.063 ^de^	3.00 ± 0.062 ^bc^
	Postoperative (immediately after the operation)	3.07 ± 0.062 ^ab^	2.98 ± 0.061 ^bc^	3.22 ± 0.054 ^a^
	Postoperative (at least one year after the operation)	2.76 ± 0.077 ^cde^	2.78 ± 0.081 ^bcde^	3.04 ± 0.065 ^ab^

Legend: PVH—Posterior vertebral height; AVH—Anterior vertebral height. Group One—pedicle stabilization using Schanz screw constructs (n = 37); Group Two—stabilization above and below the fractured vertebra using pedicle screws (n = 32); Group Three—intermediate fixation with a pedicle screw additionally inserted into the fractured vertebra (n = 45). Data are mean values ± standard error of the mean (whiskers). ^a,b,c,d,e^—significant difference: *p* < 0.05 (Tukey’s HSD test).

## Data Availability

Data available on request due to restrictions (due to privacy reasons).

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
