# Peer review of "Comparison of Methods for Short-Segment Posterior Stabilization of Lumbar Spine Fractures and Thoracolumbar Junction"

_jcm, 2024, doi:10.3390/jcm13237318_

Round 1

Reviewer 1 Report

Comments and Suggestions for Authors

Authors present a study on comparison of three stabilization techniques in 114 patients treated for thoracolumbar fractures - stabilization up/below the fracture, with Schwanz contruct and with stabilization including fractured vertebra. Superior outcomes for Schwanz screws were reported. English language editing is needed. Abstract is too short without any statistical data. Introduction needs expansion and explanation of the term Schwanz construct and Schwanz screws. One important technique is not even discussed: corpectomy of the fractured vertebra and expandable cage. Low number of patients is drawback as well as different fracture types in AO terms potentialy requiring different surgical approach. How many patients received decompressive surgery due to posterior element involvement? Authors fail to provide classification of the fractures, since not every type of fracture is suitable for every type of surgery - there is a risk of bias in comparison of spinal constructs in this way and authors should explain how did they avoid the bias here. Authors need to explain why are Groups 1 and 3 biomechanically that different that it requires comparison? All of these aspects need to go into discussion too. 

Author Response

Reviewer 1

Thank you very much for your insightful comments regarding our manuscript as they are essential for transparency and validity of our findings. We have taken your comments into considerations and did our best to explain our way of thinking or answer your questions.

  1. Abstract is too short without any statistical data.

Answer:

The methods and results sections have been corrected in the abstract.

Lines: 31-32, 40-43

  1. Introduction needs expansion and explanation of the term Schwanz construct and Schanz screws. One important technique is not even discussed: corpectomy of the fractured vertebra and expandable cage.

Answer:

In short-segmental stabilization, three different methods are used for transphyseal fixation of fractures of the lumbar spine and thoracolumbar junction Th12-L1. The first method involves stabilizing one segment above and below the fracture level. The second method involves additional stabilization of the fractured vertebra using indirect Schanz screws, which are specialized screws that provide strong anchorage for external fixation constructs, known as Schanz constructs. These constructs enable the distribution of forces along the stabilized segments, helping to maintain vertebral alignment [7]. The third short-section method involves the use of Schanz screws for pedicle stabilization.

An additional technique for severe fractures, often overlooked, is corpectomy. In this procedure, the fractured vertebra is removed and replaced with an expandable cage. This cage restores vertebral height and supports alignment, providing stabilization in cases where other surgical approaches may not be sufficient. The expandable cage can adjust to the patient’s specific anatomical needs, promoting better recovery outcomes and spinal alignment stability.

Lines: 87-92, 101-108

  1. How many patients received decompressive surgery due to posterior element involvement?

Answer:

Decompression was performed in 14, 15, and 9 patients in group one, group two, and group three, respectively. Table 2

  1. Authors fail to provide classification of the fractures, since not every type of fracture is suitable for every type of surgery - there is a risk of bias in comparison of spinal constructs in this way and authors should explain how did they avoid the bias here.

Answer:

AO spine classification was used by a qualified spine surgeon in order to evaluate the fractures and decide which operative technique is best for the patient.

The AO Spine Classification divides spinal injuries into three primary types:

  1. Type A (Compression Injuries): These involve compression fractures without significant disruption to the posterior elements of the spine. They are generally the least severe and include injuries such as wedge fractures, where the front part of the vertebral body collapses. Treatment for Type A fractures often includes conservative management or minimally invasive surgical options.
  2. Type B (Distraction Injuries): These injuries involve the posterior ligamentous complex and often result from flexion-distraction forces. They can lead to instability in the spine due to stretching or tearing of ligaments and may require surgical stabilization to prevent further displacement.
  3. Type C (Translation/Rotational Injuries): These are the most severe injuries and involve significant rotational or translational displacement, resulting in spinal instability. Type C injuries often indicate a complete dislocation of the vertebral segments. These fractures typically require surgical intervention to restore stability and protect neurological structures.

Lines: 245-251

  1. Authors need to explain why are Groups 1 and 3 biomechanically that different that it requires comparison?

Answer:

The differences between the three short-segment stabilization techniques have been described in the discussion (lines 491-557). The main difference between the first and third groups is that in the third group, Shanz screws were used for stabilization, and additionally, intermediate screws were inserted into the fractured vertebra.

In assessing the biomechanics of the spine post-stabilization, evaluations include the restoration of the proper Cobb angle, assessment of range of motion (ROM), and stiffness. In our study, we used the measurement of the Cobb angle to assess the biomechanics of the spine. Our study showed that stabilization using Shanz screws was more frequently applied in patients with higher kyphosis, indicating significant changes in the Cobb angle compared to intermediate stabilization. According to the literature, this technique allows for better restoration of proper spinal biomechanics by more precisely restoring the correct kyphosis, but it is associated with higher stiffness.

Lines: 523-594

Reviewer 2 Report

Comments and Suggestions for Authors

Thank you for the opportunity to review your manuscript titled “Comparison of Methods for Short-Segment Posterior Stabilization of Lumbar Spine Fractures and Thoracolumbar Junction.” I appreciate the chance to contribute to enhancing the clarity and rigor of your study.

Your study addresses an important topic in spinal injury treatment, comparing three stabilization techniques. I commend your efforts to provide insights into different surgical approaches and their clinical outcomes. Below, I offer some suggestions and clarifications that may strengthen your study, particularly concerning the study design, randomization, and blinding aspects. I also suggest incorporating and discussing relevant recent studies to further contextualize your findings.

1. Study Design and Randomization

The manuscript would benefit from explicitly acknowledging its quasi-experimental nature, as there is no randomization in assigning patients to treatment groups. In this study, participants were grouped based on the surgical technique selected, likely according to clinical criteria or surgeon discretion. Highlighting this in the methodology and limitations sections will guide readers in interpreting the study outcomes appropriately and help mitigate any perceived selection bias. You may consider language such as: "This study is designed as a quasi-experimental, non-randomized analysis, where group allocation was based on the surgical technique applied, reflecting real-world clinical decision-making."

2. Blinding

The lack of blinding in both participant and evaluator groups is another essential factor affecting the study’s internal validity. Unblinded designs can lead to measurement biases, particularly in self-reported outcomes such as pain and quality of life. It would be valuable to address this limitation in the discussion, noting its potential influence on outcome assessments. You may consider language such as: "As neither patients nor evaluators were blinded to the interventions, there is a potential for observational bias, particularly in subjective assessments such as pain and disability scores."

3. Statistical Analysis and Group Comparisons

Your statistical analysis was thorough; however, given the non-randomized design, it may be helpful to briefly discuss how baseline differences were handled or minimized. For example, if any demographic or baseline characteristics differed across groups, mentioning adjustments or limitations in interpreting these differences could improve the study’s transparency.

4. Terminology and Methodological Clarity

For improved clarity and alignment with standard guidelines, it might help to define the primary and secondary outcomes more explicitly. Additionally, describing the procedural nuances for each surgical technique in further detail will enhance reproducibility.

5. Limitations Section

Consider expanding the limitations section to include how the quasi-experimental design and lack of blinding may impact generalizability and internal validity. This discussion could be enriched by suggesting future randomized studies or blinding procedures to strengthen evidence in this area. 

Suggested Studies for Contextualization

Incorporating recent relevant studies could provide a broader context for your findings. Consider discussing the following works:

  1.  DOI: 10.3390/jcm12206478 – This study investigates therapeutic exercise for back pain patients and could complement your discussion of postoperative care strategies, specifically by highlighting exercise’s role in managing chronic pain and improving outcomes.

  2.  DOI: 10.1097/PHM.0000000000002239 – This umbrella review and meta-analysis on orthopedic manual therapy’s effects on chronic musculoskeletal pain sensitization would enrich the context of your study’s pain management strategies, particularly concerning approaches that address pain sensitization in musculoskeletal injuries.

  3.  

These adjustments and suggested references will not only enhance the clarity and rigor of the study but will also provide readers with a well-rounded understanding of the strengths and limitations of your findings.

Thank you again for the opportunity to review this important work.

Sincerely,

Author Response

Rewiever 2

Thank you very much for your insightful comments regarding our manuscript as they are essential for transparency and validity of our findings. We have taken your comments into considerations and did our best to explain our way of thinking or answer your questions.

  1. Study Design and Randomization

The manuscript would benefit from explicitly acknowledging its quasi-experimental nature, as there is no randomization in assigning patients to treatment groups. In this study, participants were grouped based on the surgical technique selected, likely according to clinical criteria or surgeon discretion. Highlighting this in the methodology and limitations sections will guide readers in interpreting the study outcomes appropriately and help mitigate any perceived selection bias. You may consider language such as: "This study is designed as a quasi-experimental, non-randomized analysis, where group allocation was based on the surgical technique applied, reflecting real-world clinical decision-making."

Answer:

The non-randomized, prospective study was conducted on 114 patients divided into groups based on the surgical technique selected.

Given the non-randomized design of our study, we recognize that baseline differences between groups could introduce potential biases. However, our comparison of demographic and clinical characteristics revealed no significant differences across groups, minimizing the impact of this concern on our findings. Despite this, we acknowledge that residual confounding may still exist and caution against over-interpreting causality from these observational results. We added this comment to the discussion as a limiting factor for our study.

Lines: 31-32

  1. Blinding

The lack of blinding in both participant and evaluator groups is another essential factor affecting the study’s internal validity. Unblinded designs can lead to measurement biases, particularly in self-reported outcomes such as pain and quality of life. It would be valuable to address this limitation in the discussion, noting its potential influence on outcome assessments. You may consider language such as: "As neither patients nor evaluators were blinded to the interventions, there is a potential for observational bias, particularly in subjective assessments such as pain and disability scores."

Answer:

Thank you very much for your remarks. A statement regarding the lack of blinding in the quality-of-life assessment was added to the discussion (lines 864-886).

However, in our opinion, blinding in our study was not feasible, as the surgical treatment was individually tailored to each patient by an experienced surgeon, based on the patient’s clinical and subjective assessment and a discussion about the expected surgical outcomes. Furthermore, prior to the procedure, the surgeon must plan the surgery to minimize the risk of postoperative complications. According to Polish law, the patient must also sign an informed consent form for the procedure. Before signing, the patient must be informed about the procedure, the available treatment alternatives, and the potential risks associated with the surgery.

Lines:

  1. Statistical Analysis and Group Comparisons

Your statistical analysis was thorough; however, given the non-randomized design, it may be helpful to briefly discuss how baseline differences were handled or minimized. For example, if any demographic or baseline characteristics differed across groups, mentioning adjustments or limitations in interpreting these differences could improve the study’s transparency.

Answer:

In our study, we recognized that the non-randomized design could lead to potential

confounding due to baseline differences between the groups. To mitigate this issue, we

provided a comprehensive overview of baseline characteristics for each group, including

demographic variables such as age, gender, and relevant clinical characteristics. This

descriptive analysis allowed us to identify any notable differences that could influence the

outcomes. In our study, we conducted detailed comparisons of demographic and clinical

baseline characteristics across groups, and no statistically significant differences were found.

This suggests that the groups were well-matched on these key variables, which minimizes

concerns about baseline imbalances affecting our results. Given these findings, we did not

apply additional adjustments for baseline characteristics in the primary analyses. However, we acknowledge that residual confounding remains a potential limitation and have discussed this in our limitations section to ensure transparency regarding the interpretation of our results.

  1. Terminology and Methodological Clarity

For improved clarity and alignment with standard guidelines, it might help to define the primary and secondary outcomes more explicitly. Additionally, describing the procedural nuances for each surgical technique in further detail will enhance reproducibility.

Answer:

The primary objective in treating injuries to the lumbar spine and thoracolumbar junction is to achieve a stable and lasting reduction of the fracture, as this stability is crucial for promoting the patient's rapid mobilization and return to functional activity. By securely stabilizing the fracture, early mobilization can be achieved, reducing the risks associated with prolonged immobility, such as muscle atrophy, decreased cardiovascular conditioning, and potential complications like pressure ulcers. Restoration of the normal sagittal alignment of the spine, while essential for long-term functional outcomes and reducing the risk of chronic pain or deformity, is considered a secondary goal. Correct sagittal alignment supports overall spinal health and load distribution, but immediate stability and mobilization take precedence to optimize recovery.

Lines: 523-533

  1. Limitations Section

Consider expanding the limitations section to include how the quasi-experimental design and lack of blinding may impact generalizability and internal validity. This discussion could be enriched by suggesting future randomized studies or blinding procedures to strengthen evidence in this area. 

Answer:

A paragraph about the limitations of our work was added to the discussion - Lines: 872-886.  A sentence was also added about future work being randomized – lines: 903-904

Round 2

Reviewer 1 Report

Comments and Suggestions for Authors

Authors have sufficiently responded to remarks.

Author Response

Authors have sufficiently responded to remarks.

Thank you very much again for you insightful comments. We think improving the aforementioned points impacted the clarity and design of our manuscript. 

Reviewer 2 Report

Comments and Suggestions for Authors

Thank you for the opportunity to review the revised manuscript. I appreciate the effort the authors have put into addressing the comments and suggestions provided during the initial review. The improvements are evident in several areas, particularly in clarifying the study design, elaborating on statistical analyses, and addressing some of the study's limitations. However, there remain aspects that could benefit from further refinement to enhance the rigor and transparency of the work. Below, I outline my observations and suggestions for additional revisions.

Firstly, I recognize the authors' efforts to clarify the quasi-experimental nature of the study and to address the issue of randomization. The revised methodology explicitly acknowledges the lack of randomization and explains that group allocation was based on clinical decision-making. This is a welcome addition. However, the discussion around "residual confounding" remains somewhat vague. It would strengthen the manuscript to include concrete examples of potential confounders that might influence the outcomes, even if the baseline characteristics were statistically similar across groups. A deeper exploration of these potential confounders would provide readers with a clearer understanding of the study's limitations.

Regarding the lack of blinding, I note that the authors have included a statement in the discussion, acknowledging this as a limitation and explaining why blinding was not feasible. While this explanation is detailed, it leans heavily toward justifying the absence of blinding rather than focusing on its implications for the study's findings. A more concise acknowledgment of the limitations imposed by the lack of blinding, alongside a discussion of its potential impact on subjective measures like pain and quality of life, would better serve the transparency of the manuscript. Additionally, suggesting how future studies might incorporate blinding where feasible could add value to the discussion.

The section on statistical analysis has been improved, with the authors providing a thorough explanation of how baseline differences were evaluated and managed. This enhances the study’s credibility. However, this information would be more impactful if explicitly integrated into the limitations section to highlight its relevance to the study's validity. Making this connection clear would provide a more balanced perspective on the study’s strengths and weaknesses.

In terms of methodological clarity, the revised manuscript has made progress in defining the primary and secondary outcomes and in elaborating on the procedural details of the surgical techniques. However, these details remain somewhat scattered across the manuscript. A dedicated subsection that consolidates all procedural information would significantly improve the clarity and reproducibility of the study.

The limitations section has been expanded, which is commendable. The authors now address the quasi-experimental design and the lack of blinding. However, the discussion on generalizability remains cursory. For instance, it would be beneficial to explore how the findings might apply—or fail to apply—to different patient populations or clinical settings. This additional detail would provide readers with a clearer picture of the broader implications of the study.

Finally, I noticed that the suggested studies for contextualization, specifically those addressing therapeutic exercise and pain sensitization (DOIs 10.3390/jcm12206478 and 10.1097/PHM.0000000000002239), do not appear to have been incorporated. Including these references in the discussion would not only enrich the context of the findings but also demonstrate a comprehensive engagement with the existing literature. For example, these studies could enhance the discussion on postoperative care strategies and the management of chronic pain, which are relevant to the manuscript’s focus.

Overall, I commend the authors for their diligent work in revising the manuscript and addressing many of the initial concerns. I encourage them to make the additional refinements outlined above to further strengthen the study. Thank you again for the opportunity to contribute to the development of this important work.

Author Response

Thank you very much for your insightful comments. We did our best to answer them to the best of our knowledge.

  1. Firstly, I recognize the authors' efforts to clarify the quasi-experimental nature of the study and to address the issue of randomization. The revised methodology explicitly acknowledges the lack of randomization and explains that group allocation was based on clinical decision-making. This is a welcome addition. However, the discussion around "residual confounding" remains somewhat vague. It would strengthen the manuscript to include concrete examples of potential confounders that might influence the outcomes, even if the baseline characteristics were statistically similar across groups. A deeper exploration of these potential confounders would provide readers with a clearer understanding of the study's limitations.

Response:

Thank you for your thoughtful comments and for highlighting the importance of addressing potential residual confounding in our study. While we recognize the value of conducting a sensitivity analysis to assess the potential impact of unmeasured confounders, we were unfortunately unable to perform such an analysis due to data limitations. Specifically, certain variables, such as detailed rehabilitation protocols, bone density, and comorbidities, were not systematically collected during the study.

To address this limitation, we have carefully outlined the potential sources of residual confounding in the revised discussion section (lines: 845-869) and acknowledged how these might influence our findings. While we could not directly evaluate the impact of these factors, we sought to mitigate their influence by ensuring baseline comparability across groups and employing statistical adjustments for measured confounders.

We also emphasize the need for future studies, including randomized controlled trials or those employing advanced statistical methods like propensity score matching, to further validate our findings and minimize confounding. We have incorporated these points into the discussion to provide a transparent account of our study's limitations.

  1. Regarding the lack of blinding, I note that the authors have included a statement in the discussion, acknowledging this as a limitation and explaining why blinding was not feasible. While this explanation is detailed, it leans heavily toward justifying the absence of blinding rather than focusing on its implications for the study's findings. A more concise acknowledgment of the limitations imposed by the lack of blinding, alongside a discussion of its potential impact on subjective measures like pain and quality of life, would better serve the transparency of the manuscript. Additionally, suggesting how future studies might incorporate blinding where feasible could add value to the discussion.

Response:

We greatly appreciate your insightful comment. This has encouraged us to broaden our literature review, particularly focusing on works such as DOI: 10.12688/f1000research. While blinding of patients is possible, it is generally only applicable in clinical trials, not in retrospective or prospective studies. Additionally, blinding of the assessor can be applied in all types of studies, but it should be noted that blinding of the surgeon is not feasible, as the surgeon must perform the specific procedure.

In response to your point on the potential impact of the lack of blinding, we have added information regarding meta-analyses by Moustgaard et al. (2020) and Efficace et al. (2022), which found no significant differences between studies with and without blinding. However, since no such research exists for assessments of quality of life following short-segment spinal stabilization, we are unable to determine how the lack of blinding might have influenced the results. We agree with your suggestion and recommend that future studies in this area incorporate randomization and blinding procedures to address this limitation. These details have been incorporated into the discussion section of the manuscript – lines 870-882.

  1. The section on statistical analysis has been improved, with the authors providing a thorough explanation of how baseline differences were evaluated and managed. This enhances the study’s credibility. However, this information would be more impactful if explicitly integrated into the limitations section to highlight its relevance to the study's validity. Making this connection clear would provide a more balanced perspective on the study’s strengths and weaknesses.

Response:

Thank you for your thoughtful feedback and recognition of the improvements made to the statistical analysis section. In response to your suggestion, we have integrated the discussion on baseline differences and their evaluation into the limitations section. This addition emphasizes the relevance of these analyses to the study's validity and provides a balanced perspective on its strengths and weaknesses.

Furthermore, the methodology section has been revised to explicitly describe the steps taken to assess baseline differences between groups, including the specific variables compared (e.g., sex distribution, hospital stay duration, and injury causes) and the statistical methods employed (e.g., chi-square test, Tukey’s HSD) (Lines:248-254). These revisions aim to improve the clarity and comprehensiveness of the manuscript, further enhancing its credibility.

Thank you again for your valuable feedback, which has been instrumental in refining the presentation of our study.

  1. In terms of methodological clarity, the revised manuscript has made progress in defining the primary and secondary outcomes and in elaborating on the procedural details of the surgical techniques. However, these details remain somewhat scattered across the manuscript. A dedicated subsection that consolidates all procedural information would significantly improve the clarity and reproducibility of the study.

Response:

Thank you very much for your constructive feedback, which has allowed us to improve our manuscript.

A section titled "Surgical Techniques and Procedures"(Lines: 135-195) has been added to the Materials and Methods section. This section includes all the information regarding the number and distribution of patients, a detailed description of the surgical procedures, an explanation of the radiological assessment, and an evaluation of patients' quality of life.

  1. The limitations section has been expanded, which is commendable. The authors now address the quasi-experimental design and the lack of blinding. However, the discussion on generalizability remains cursory. For instance, it would be beneficial to explore how the findings might apply—or fail to apply—to different patient populations or clinical settings. This additional detail would provide readers with a clearer picture of the broader implications of the study.

Response:

Thank you for your valid point. The limitations section has been further expanded (Lines: 885-886) to include suggestions from this review. Our study showed that regardless of the fixation technique chosen, the correct curvature of the spinal column can be significantly restored, although no significant differences between the groups were observed in the postoperative follow-up examination. However, it should be noted that the highest post-traumatic kyphosis was observed in patients in the Schanz group, which proves that this technique is preferred in patients with a large Cobb angle [11]. It was also observed that the Schanz technique ensured a better recovery of AVH. As well as the assessment of the quality of life of patients after the operation using the WAS and ODI scales showed that the stabilization using Shanz screws is significantly better. Therefore, it can be stated that in medical practice this may be the best type of short-segment stabilization of lumbar spine and thoracolumbar junction injuries. Especially in patients with high fracture kyphotization. However, the lack of analysis of confounding factors and blinding of the study may have influenced the results. Our study was also limited to one hospital, one patient population, and one surgical team performing the procedures. However, the lack of analysis of confounding factors and the blinding of the study may have influenced the obtained results. Also, our study was limited to one hospital, one patient population and one surgical team performing the procedures. We therefore recommend further studies in this direction to confirm this hypothesis. However, our team recommends further studies in this direction based on blinding and randomization to obtain more reliable results. We also recommend that further studies be conducted in several centers to check whether the obtained results are influenced by the patient and can be reproduced by other surgical teams.

  1. Finally, I noticed that the suggested studies for contextualization, specifically those addressing therapeutic exercise and pain sensitization (DOIs 10.3390/jcm12206478 and 10.1097/PHM.0000000000002239), do not appear to have been incorporated. Including these references in the discussion would not only enrich the context of the findings but also demonstrate a comprehensive engagement with the existing literature. For example, these studies could enhance the discussion on postoperative care strategies and the management of chronic pain, which are relevant to the manuscript’s focus.

Response:

Thank you for your input. Paragraph added to materials and methods, additionally, included study limitation regarding lack of control on postoperative care in discussion. The suggested studies have also been added and cited in the manuscript.

“Postoperative care

Weeks following the surgery, physiotherapy played a crucial role in the patient’s recovery [10,11]. During the initial six weeks, the patient performed light activities to improve circulation and prevent stiffness, while protecting the surgical site. Range-of-motion exercises for the hips and lower extremities were included, along with short, supervised walking sessions. Core engagement exercises were cautiously introduced to maintain stability.

In the later weeks, the rehabilitation program advanced to core strength, flexibility, and endurance exercises. The patient engaged in light strengthening activities for the back, abdominal, and pelvic muscles while gradually increasing walking distances and participating in controlled, low-impact activities like swimming or stationary cycling.

After 12 weeks, the focus shifted to restoring full function and preparing the patient for daily activities and work. Advanced strengthening and flexibility exercises were implemented, emphasizing posture correction and proper body mechanics to protect the spine during movements. The patient avoided high-impact activities or heavy lifting until cleared by the surgeon.

Throughout the program, exercises were tailored to the patient’s progress, ensuring appropriate intensity and avoiding excessive strain on the spine. Education on proper movement techniques and lifestyle adjustments was provided to support long-term recovery and prevent reinjury.”

Lines: 196-215, 887-894
